# Recent Advances in Enzyme-Nanostructure Biocatalysts with Enhanced Activity

**Jing An [1], Galong Li [1,\*], Yifan Zhang [1], Tingbin Zhang [2], Xiaoli Liu [1], Fei Gao [1], Mingli Peng [2], Yuan He [2]**  **and Haiming Fan [1,2,\*]**

[1] School of Chemical Engineering & Key Laboratory of Resource Biology and Biotechnology in Western China, Ministry of Education, The College of Life Sciences, Northwest University, Xi'an 710069, China; anjing@stumail.nwu.edu.cn (J.A.); zhangyf@stumail.nwu.edu.cn (Y.Z.); liuxiaoli19870108@126.com (X.L.); gaofei@stumail.nwu.edu.cn (F.G.)

[2] Key Laboratory of Synthetic and Natural Functional Molecule Chemistry of the Ministry of Education, College of Chemistry and Materials Science, Northwest University, Xi'an 710069, China; zhangtb@nwu.edu.cn (T.Z.); mlpeng@nwu.edu.cn (M.P.); yuanhe@nwu.edu.cn (Y.H.)

\* Correspondence: galongli@nwu.edu.cn (G.L.); fanhm@nwu.edu.cn (H.F.)

**Abstract:** Owing to their unique physicochemical properties and comparable size to biomacromolecules, functional nanostructures have served as powerful supports to construct enzyme-nanostructure biocatalysts (nanobiocatalysts). Of particular importance, recent years have witnessed the development of novel nanobiocatalysts with remarkably increased enzyme activities. This review provides a comprehensive description of recent advances in the field of nanobiocatalysts, with systematic elaboration of the underlying mechanisms of activity enhancement, including metal ion activation, electron transfer, morphology effects, mass transfer limitations, and conformation changes. The nanobiocatalysts highlighted here are expected to provide an insight into enzyme–nanostructure interaction, and provide a guideline for future design of high-efficiency nanobiocatalysts in both fundamental research and practical applications.

**Keywords:** increased activity; enzyme immobilization; nanobiocatalysts; nanomaterials; hybrid materials

---

## 1. Introduction

The pursuing of efficient supports to improve enzyme performance has been one of the most important directions in biotechnology since the first example of enzyme immobilization in 1916 [1–3]. The rapid growth in nanotechnology offers a wealth of opportunities for the successful combination of enzymes with various nanostructured materials, namely enzyme–nanostructure biocatalysts (nanobiocatalysts) [4–9]. In contrast to conventional bulk supports, nanostructured supports possess plenty of advantages, such as large specific surface area, reduced mass transfer limitation, ease of surface modifications, unique geometry and size/shape-dependent characteristics. Given these aforementioned advantages, nanobiocatalysts have shown improved stability and recyclability as well as reusability compared to free enzyme. Although immobilization is usually associated with the distortions in the enzyme structure and a decrease in enzyme activity due to the support effect and immobilization methods, nanobiocatalysts have shown a broad spectrum of applications in environmental remediation, biosensing, biomedicine, and industrial biocatalysis [8,10,11].

To attain the high catalytic activity required for nanobiocatalysts in many applications, considerable efforts have been devoted to implementing favorable interactions between enzymes and the nanostructured supports [3,10–13]. Nanobiocatalysts with increased enzyme activity can be obtained via either covalent binding, adsorption, entrapment, or encapsulation [5,6,14,15]. The structure of

nanoscale supports, such as nanoparticles, nanowires, microspheres, metal-organic frameworks, and nanoflowers, has also drawn extensive attention in recent years [5,16–22]. A great deal of effort has also been made to design nanostructured supports with a variety of components including noble metal (e.g., Au) [23,24], metal oxides (e.g., $Cu_2O$, $Fe_3O_4$, $SiO_2$, $Ti_8O_{15}$, alumina) [16,25–33], polymer (e.g., $Cu^{2+}$/PAA/PPEGA matrix, aldehyde-derived Pluronic polymer, polycaprolactone) [34–36], metal-organic frameworks (e.g., zeolitic imidazolate framework) [37–39], carbon based (e.g., carbon dots, carbon nanotubes) [40–44], and complex compounds (e.g., $Cu_3(PO_4)_2 \cdot 3H_2O$, $Ca_3(PO_4)_2$, $Co_3(PO_4)_2 \cdot 8H_2O$, $Mn_3(PO_4)_2$, $Ca_8H_2(PO_4)_6$, $Cu_4(OH)_6)SO_4$, $CaHPO_4$, $Zn_3(PO_4)_2$, Mg-Al layered double hydroxide, CdSe/ZnS quantum dots) [17,45–59]. These representative supports, which possess unique chemical and physical properties, such as controllable release of ion activator and synergic catalysts and response to external stimuli, are able to regulate the enzyme-support interaction and eventually lead to an unprecedented enhancement in immobilized enzyme activity [7,60,61].

Overall, recent years have witnessed great success in the interactions between artificial nanostructured supports and natural enzymes for enhanced activity. This review will focus on recent advances in this field in the past 10 years, with an emphasis on various mechanisms behind the boosted catalytic activities, including reduced mass transfer limitation, interfacial ion activation, local heating effect, synergistic effects, conformational changes, substrate channeling and so on. A better understanding of the relationship between the characteristics of the nanostructured supports and the enhanced activities of nanobiocatalysts, may provide new insights in designing efficient nanobiocatalysts for various applications.

## 2. Mechanisms behind Enhanced Activities of Nanobiocatalysts

An overview of recently reported nanobiocatalysts with increased activities is listed in Table 1. Among these, the activity enhancement mechanism is attributed to a favorable interaction between the enzyme and the nanostructured support, which can be roughly categorized into the following strategies: metal ion activation, morphological effects, temperature effects, enhanced electron transfer, conformational modulation, and multi-enzyme cascade reaction (Figure 1). Representative work on each aspect will be discussed here.

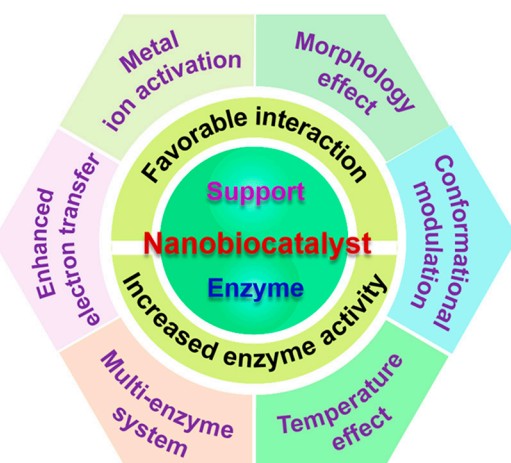

**Figure 1.** Design of nanobiocatalysts with favorable interactions between enzymes and nanostructured supports for enhanced activities.

**Table 1.** Recent nanobiocatalysts with increased activities, constructed by various nanostructure supports and enzymes.

| Enzymes | Supports | Increased Activities (Folds) | Ref. |
|---|---|---|---|
| Laccase | $Cu_3(PO_4)_2$ nanoflower | 6.50 | [48] |
| Horseradish peroxidise | $Cu_3(PO_4)_2$ nanoflower | 5.06 | [62] |
| Laccase | $Cu_3(PO_4)_2$ nanoflower | 1.50 | [49] |
| Lipase | Polycaprolactone nanofiber | 14.00 | [36] |
| Laccase | Au nanoparticle | 1.91 | [23] |
| Laccase | Carbon dots | 1.92 | [41] |
| Laccase | $Cu^{2+}$/PAA/PPEGA | 4.47 | [34] |
| Laccase | Single-walled carbon nanotube | 6.00 | [42] |
| Laccase | $Cu_3(PO_4)_2$ hybrid microsphere | 3.60 | [50] |
| Laccase | Membrane/nanoflower | 2.00 | [63] |
| Laccase | Mesoporous silica nanoparticle | 1.20 | [64] |
| Laccase | $Cu_2O$ nanoparticle | 4.00 | [25] |
| α-amylase | $CaHPO_4$ nanoflower | 37.5 | [17] |
| β-galactosidase | Mg-Al layered double hydroxide | 30.00 | [58] |
| α-chymotrypsin | $Ca_3(PO_4)_2$ nanoflower | 2.66 | [52] |
| Horseradish peroxidase, Glucose oxidase | $Cu_3(PO_4)_2·3H_2O$ nanocrystal | 1.40 3.10 | [65] |
| Cytochrome c | ZIF-8 metal-organic framework | 10.00 | [37] |
| Lipase, Cytochrome c | Pluronic polymer | 67.00, 670.0 | [35] |
| L-2-HAD$_{ST}$ dehalogenase | $Fe_3O_4$ nanoparticles/hydrogel | 2.00 | [26] |
| Lipase | $Cu_3(PO_4)_2$ nanoflower | 4.60 | [66] |
| Organophosphorus hydrolase | $Co_3(PO_4)_2·8H_2O$ nanocrystal | 3.00 | [53] |
| Amylase, Cellulase, Lipase | $Ti_8O_{15}$ nanoparticle | 13.00, 5.00, 12.00 | [32] |
| Glucose oxidase | CdSe/ZnS quantum dot | 2.00 | [59] |
| Carbonic anhydrase | $Cu_3(PO_4)_2$ nanoflower, $Ca_8H_2(PO_4)_6$ nanoflower | 2.86, 1.49 | [54] |
| D-psicose 3-epimerase | $Co_3(PO_4)_2$ nanoflower | 7.20 | [67] |
| Lipase | Carbon nanotube, $Cu_3(PO_4)_2$ nanoflower | 68.00, 51.00 | [40] |
| Laccase | $Cu_2O$ nanowire mesocystal | 10.00 | [16] |
| Laccase | $Cu(OH)_2$ nanocage | 18.00 | [68] |
| β-galactosidase | $Fe_3O_4$ nanoring | 1.80 | [27] |
| Lipase | Polyacrylamide nanogel | 2.00 | [69] |
| Lipase | Pluronic polymer | 11.00 | [70] |
| Lipase | siliceous mesocellular foam | 25.00 | [71] |
| Horseradish peroxidase | DNA scaffold | >3.00 | [72] |
| Horseradish peroxidase | Magnetic nanoparticle | 10.00 | [28] |
| Cytochrome c | Copper hydroxysulfate | 143.00 | [55] |
| Laccase | $Fe_3O_4$-NH$_2$-PEI $Fe_3O_4$-NH$_2$ | 101.33 74.45 | [30] |
| Glucose oxidase | Anodic alumina nanochannel | 80.00 | [33] |
| Lipase | $Zn_3(PO_4)_2$ hybrid nanoflower | 1.47 | [56] |
| Invertase | $CaHPO_4$ hybrid nanoflower | 2.03 | [57] |
| Urease | $Cu_3(PO_4)_2·3H_2O$ nanoflower | 40.00 | [73] |
| L-arabinitol 4-dehydrogenase, NADH oxidase | $Cu_3(PO_4)_2·3H_2O$ nanoflower | 2.46, 1.44 | [74] |
| Laccase | Copper alginate | 3.00 | [75] |
| Hydroxylase | $Cu_3(PO_4)_2·3H_2O$ nanoflower | 1.62 | [76] |
| Pyruvate kinase/lactate dehydrogenase | Semiconductor quantum dot | >50.00 | [77] |
| Alkaline protease | Hollow silica nanosphere | 2.40 | [78] |

## 2.1. Morphology Effect of Nanoscale Support

The morphology of nanoscale supports has a great impact on the enhancement of enzyme activities and the stability of nanobiocatalysts (Figure 2) [12]. Continuous efforts have yielded many nanostructured supports with various morphologies to provide a large surface area for increased enzyme loading and reduced diffusion resistance [21,79]. Many enzyme immobilization strategies have been developed in consideration of the optimal morphology of supports for minimizing the effect of conformation distortions and the deactivation of immobilized enzymes. As shown in Figure 2, nanobiocatalysts with increased activity are constructed through insertion of nanoparticles into enzyme molecules [23], immobilization of enzymes on the surface of nanoparticle/nanorod/2-dimensional nanomaterials [27,28,34], encapsulation of enzymes in porous supports [37,39], embedding/enveloping enzymes on/in 3-dimensional nanostructured supports [16,48,80]. Horseradish peroxidase was immobilized on a nanoscale DNA scaffold with three addressable sites [72]. The enhanced enzyme activity (>300%) of this nanobiocatalyst was ascribed to preferential binding of the substrates to the minor groove of the double DNA helix. Cytochrome c was encapsulated into spindle-like copper hydroxysulfate nanocrystals, which exhibited a 143-fold increase in catalytic efficiencies ($k_{cat}/K_m$) [55]. The spindle-like nanostructured support with rough surface provided large specific surface area, resulting in an apparent decrease in $K_m$. Therefore, more substances can be accumulated around and within the nanostructured supports for efficient catalytic reactions. Jiang et al. immobilized glucose oxidase on a porous anodic alumina nanochannel membrane [33]. In this case, the $O_2$ molecules can easily go through the nanochannels and participate in the catalytic reaction in an aqueous solution, giving rise to a significant increase in catalytic efficiency (by 80-fold). The construction of a gas-nanochannel-liquid system was shown to boost gas-involving reactions, which introduces a brand-new idea to enhance the activity of immobilized enzymes using gas substrates. Manganese peroxidase was encapsulated in vault nanoparticles, by which the obtained nanobiocatalysts exhibited an enhanced phenol degradation capability three times larger than that of the unpackaged enzymes [80]. The vault nanoparticles with hollow structures were also reported to contribute to an increased catalytic activity of the encapsulated enzymes towards harsh conditions.

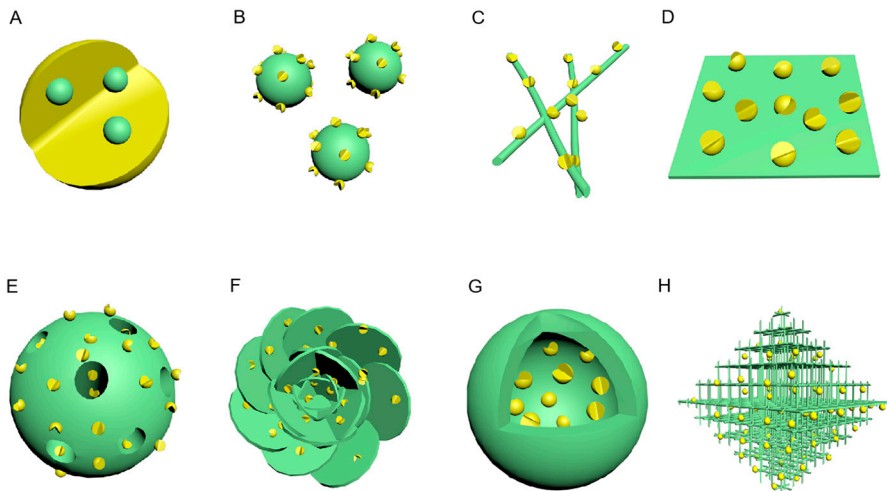

**Figure 2.** Representative morphologies of nanobiocatalysts with increased activities. (**A**) Insertion of nanoparticles into enzyme molecules. (**B–D**) Immobilization of enzymes on the surface of nanoparticle/nanorod/2 dimensional nanomaterial. (**E**) Encapsulation of enzymes in porous supports. (**F–H**) Embedding/enveloping enzymes on/in 3-dimensional nanostructured supports.

Zeng et al enveloped α-amylase on nanoflowers, nanoplates, and parallel hexahedrons (Figure 3A–C), with their reaction rate constants (*k*) reported to be $16.5 \times 10^{-3}$, $8.0 \times 10^{-3}$, and $1.2 \times 10^{-3}$ s$^{-1}$, respectively [17]. It was shown that the α-amylase-nanoflowers possess the highest

catalytic activity, likely due to their higher surface-to-volume ratios and dramatically reduced mass transfer limitations, as enzymes rested on the surface of a flower petal have much more possibility to react with the substrate. While this is promising, the presence of dead areas in these nanoflowers may hinder their activity enhancement. For a fair comparison of the catalytic activities of various nanobiocatalysts, Fan et al. proposed the use of "specific nanobiocatalyst activity" [16]. The specific enzyme activity is calculated by A = U/P, where P is mg of enzyme. One unit of enzyme activity (U) is the amount of enzyme required to oxidize 1 µmol substrate per minute (or second). The specific nanobiocatalyst activity is defined as A' = U/B, where B is mg of both enzyme and support. It was reported that the immobilized enzymes with a high specific enzyme activity will actually have a low specific nanobiocatalyst activity. In this work, laccae–$Cu_2O$ nanowire mesocrystal hybrid materials exhibited a 2.2-fold increase in specific nanobiocatalyst activity, the highest among the previously reported examples. The nanowire mesocrystal supports display an open octahedra morphology, which is assembled using well-organized anisotropic nanowires as building blocks (Figure 2H). Compared with the compact nanostructured supports, the nanowire mesocrystal has a much larger surface-to-volume ratio and interpenetrating inner channels, which significantly minimize mass transfer limitation (Figure 3F). The 3D ordered spatial arrangement of laccases (Figure 2H) eliminated the phenomena of overlapped nanostructures and provided abundant accessible active sites to substances. In comparison, the laccase–nanowire mesocrystal system attained the highest specific enzyme activities, about 10 and 8 times higher than that of laccase–nanocube and laccase–nanowire systems, respectively (Figure 3D,F). The nanocubes and nanowires were considered to have a greater likelihood to form temporary and permanent aggregations. After that, the disastrous aggregation may exert serious steric hindrance and lead to a significant diffusion resistance, leading to the different diffusion pathways of the substrates, as illustrated in the insets of Figure 3D,F. However, it is worth noting that these 3-dimensional porous supports actually have some disadvantages in increasing the activity of immobilized enzymes. For instance, the immobilized enzymes inside porous supports cannot interact with external substances. The porous support may generate a pH inside the pore that may differ from the optimal pH in the bulk. The enzymes inside the pore are far from optimal pH, causing a decrease in catalytic activity.

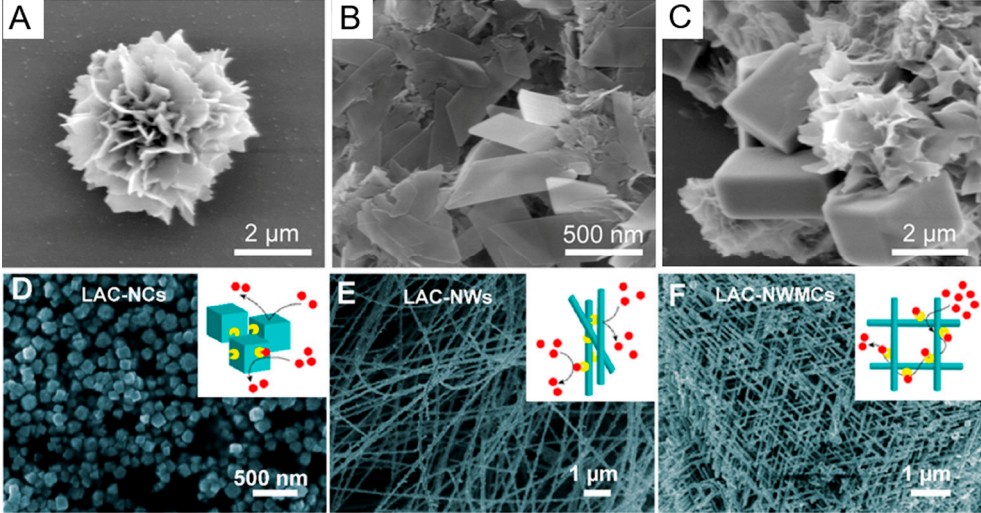

**Figure 3.** SEM images of $CaHPO_4$-α-amylase nanobiocatalysts, (**A**) nanoflowers, (**B**) nanoplates, and (**C**) parallel hexahedrons. Reconstructed with permission from Ref. [17], Copyright (2013) American Chemical Society. SEM images of $Cu_2O$–laccase nanobiocatalysts, (**D**) nanocubes, (**E**) nanowires, and (**F**) nanowire mesocrytal, insets are the schematic illustrations of the plausible substrate diffusion pathways for these hybrid materials. Reconstructed with permission from Ref. [16], copyright (2018) American Chemical Society.

## 2.2. Metal Ions Activation

Metal ions may bind to enzymes and serve as cofactors. About one third of enzymes are known to be metalloenzymes [81]. The metalloenzyme-based nanobiocatalysts have been the most studied because their activities can be activated by the presence of the corresponding metal ions. In such systems, the nanostructured supports can undergo an inherent dissolution–crystallization dynamic process in solution to release the required metal ions, including copper, cobalt and manganese. Metal ions activation of immobilized enzymes has been summarized in Table 2. Ge and the co-workers prepared a laccase-$Cu_3(PO_4)_2 \cdot 3H_2O$ hybrid nanoflower through a facile and general coprecipitation method, which exhibits a 6.5-fold increase in enzyme activity (Figure 4A) [48]. In this design, $Cu^{2+}$ ions form complexes with enzyme molecules during the crystal growth process and play an important role in the activation of laccase. Such an activation effect has also been employed to achieve a 5-fold enhancement in the activity of horseradish peroxidase–$Cu_3(PO_4)_2 \cdot 3H_2O$ hybrid nanoflowers [62]. Liu et al. covalently bound laccase on a $Cu^{2+}$ adsorbed [poly(acrylic acid)/poly(poly-(ethylene glycol) acrylate)]. The immobilized laccase exhibited both enhanced activity (4.47-fold) and thermal stability [34]. Apart from the $Cu^{2+}$ activation effect, the $Cu_2O$ nanowire mesocrystal support was found to serve as a $Cu^+/Cu^{2+}$ self-sustainable reservoir, which contributed to the remarkably increased enzyme activity, as shown in Figure 4B [16]. The enhancement in activity was mainly due to two factors: the $Cu^+$ can be incorporated into the active center of laccase and the $Cu^{2+}$ can enhance intramolecular electron transfer. Additionally, cytochrome c was incorporated into the metal-organic frameworks (ZIF-8), in which $Zn^{2+}$ was considered an important factor for the 10-fold increase in activity [37]. Taken together, the strategy of metal ion activation can provide a vital method in the designing of metalloenzyme-based nanobiocatalysts with efficient catalytic activities.

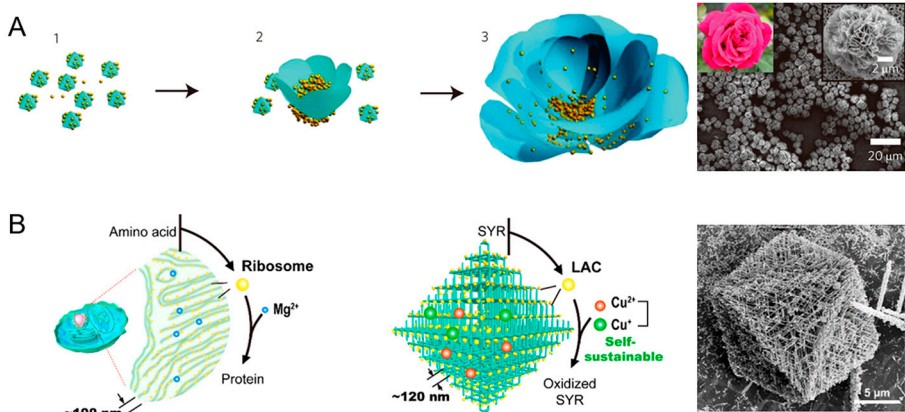

**Figure 4.** (**A**) Formation process of enzyme-incorporating $Cu_3(PO_4)_2 \cdot 3H_2O$ nanoflowers, comprising three steps: nucleation and formation of primary crystals, growth of crystals, formation of nanoflowers. Reprinted with permission from Ref. [48]. Copyright (2012) Nature Publishing Group. (**B**) Bioinspired fabrication of enzyme–nanowire mesocrystal hybrid materials by mimicking natural rough endoplasmic reticulum. Reconstructed with permission from Ref. [16], copyright (2018) American Chemical Society.

**Table 2.** The effects of metal ion and temperature on the enhanced activities of immobilized enzymes.

| Enzymes | Effects | Increased Activities (Folds) | Ref. |
|---|---|---|---|
| Laccase, carbonic anhydrase | $Cu^{2+}$ | 6.50 2.60 | [48] |
| Laccase | $Cu^{2+}$ | 4.47 | [34] |
| Laccase | $Cu^{2+}$ | 3.60 | [50] |

**Table 2.** *Cont.*

| Enzymes | Effects | Increased Activities (Folds) | Ref. |
|---|---|---|---|
| Laccase | $Cu^{2+}$ | 4.00 | [25] |
| $\alpha$-amylase | $Ca^{2+}$ (Allosteric Effect) | 37.5 | [17] |
| $\beta$-galactosidase | $Mg^{2+}$ (Allosteric Effect) | 30.00 | [58] |
| Cytochrome c | $Zn^{2+}$ | 10.00 | [37] |
| Organophosphorus hydrolase | $Co^{2+}$ (Allosteric Effect) | 3.00 | [53] |
| Carbonic anhydrase | $Cu^{2+}$, $Ca^{2+}$ | 2.86, 1.49 | [54] |
| Urease | $Cu^{2+}$ | 40.00 | [73] |
| D-psicose 3-epimerase | $Co^{2+}$ | 7.20 | [67] |
| Laccase | $Cu^{+}$ and $Cu^{2+}$ | 10.00 | [16] |
| Laccase | $Cu^{2+}$ | 18.00 | [68] |
| Lipase, Cytochrome c | Temperature responsiveness in organic solvents | 67.00, 670.0 | [35] |
| L-2-HAD$_{ST}$ dehalogenase | Magnetothermal effect | 2.00 | [26] |
| Laccase | Increased temperature by local surface plasma resonance effect | 1.91 | [23] |
| Amylase, Cellulase, Lipase | Solar-to-thermal conversion | 13.00, 5.00, 12.00 | [32] |
| $\beta$-galactosidase | Magnetothermal effect | 1.80 | [27] |
| Lipase | Temperature responsiveness in organic media | 11.00 | [70] |

### 2.3. Electron Transfer Effect

Enzymatic redox reactions are closely related to the charge transfer process, especially during oxidoreductase-catalyzed processes [42]. Conductive nanostructured supports have been demonstrated as promising candidates to enhance charge transport of redox enzymes. The representative nanomaterials include Au nanoparticles [23], CdS nanorods [82], carbon dots [41], carbon tubes [42], etc. Especially, Au nanoparticles have recently emerged as one of the most prominent supports due to excellent biocompatibility, high specific surface area, and quantum size effects. Au nanoparticles were inserted into laccase, which showed a 1.91-fold enhancement in catalytic activity. The insertion can lead to a looser protein structure and help enzymes to extract electrons from the substrates (Figure 5A) [23]. Kang et al. bound laccase with the -PO$_3$ groups on the surface of carbon dots, and found the groups can combine with the T1 Cu site of laccase, offering increased electron transfer and substrate affinity [41].

### 2.4. Temperature Effects

The rate of enzymatic reactions is known to be influenced by the temperature of the reaction solution both for general and thermophilic enzymes. Activity measurement of an immobilized enzyme at temperature above the optimal conditions may bring up a significant enhancement in enzyme activity [2]. Adjusting the temperature for improving enzymatic performances has been elaborately investigated in the past decades. Some nanostructured supports can absorb light/electromagnetic waves and convert them into heat for promoting activities of nanobiocatalysts. The effect of temperature on the enhanced activities of immobilized enzymes has been summarized in Table 2 [27,44]. Zhu et al. reported the first example of a temperature-responsive enzyme-polymer nanobiocatalyst, which showed markedly enhanced catalytic activities in organic media at 40 °C [35]. Controlled activation of non-photosensitive enzymatic reactions using a light-to-thermal strategy was also achieved with the help of photosensitive nanostructured supports such as noble metal nanoparticles and semiconductor nanomaterials [83,84]. Blankschien et al. prepared a thermophilic enzyme–Au nanorod nanobiocatalyst that showed an improvement of enzymatic reaction rate of about 60% upon photothermal activation [24].

Although encouraging achievements have been made in light activation of immobilized enzymes, it is important to protect enzymes from deactivation by excessive temperatures and photogenerated holes, as well as reactive oxygen species, especially for constructing a nanobiocatalyst using semiconductor nanomaterials as supports [85].

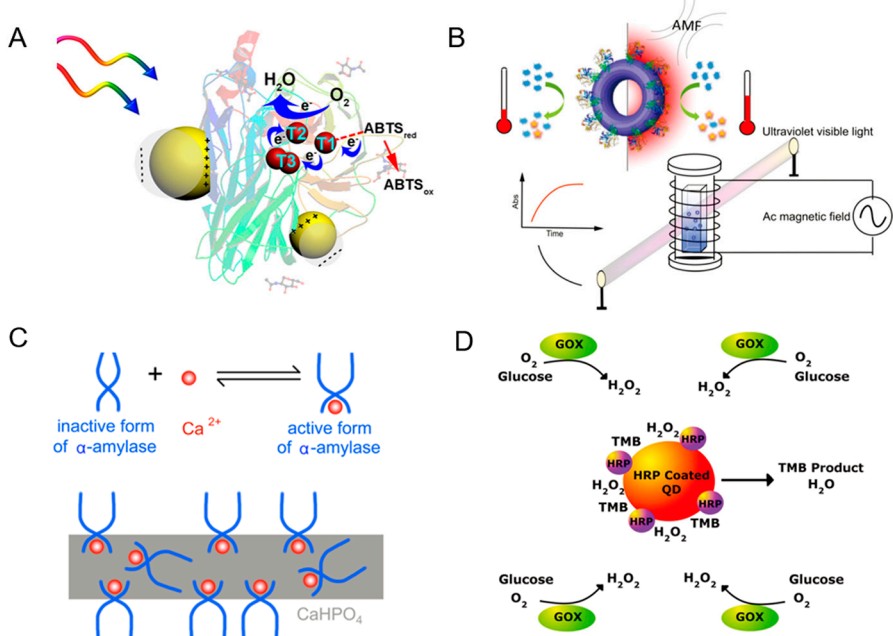

**Figure 5.** (**A**) Schematic of the Au–laccase hybrids with enhanced electron transfer. Reconstructed with permission from Ref. [23], Copyright (2015) American Chemical Society. (**B**) Illustration of the FVIO-β-Gal hybrids and the experimental set-up, the activities were tuned by AFM-triggered local heating. Reprinted with permission from Ref. [27]. Copyright (2019) RSC Pub. (**C**) The α-amylase-CaHPO$_4$ nanoflower nanobiocatalyst Ca$^{2+}$ binds to allosteric sites in inactive α-amylase and generates active α-amylase. Reproduced with permission from Ref. [17], copyright 2013, American Chemical Society. (**D**) Diagram of the GOX/HRP–CdSe/ZnS QDs system with enhanced coupled enzymatic activity. Reconstructed with permission from Ref. [59], copyright (2017) RSC Pub.

In comparison with light, magnetic fields can induce the conversion of electromagnetic energy into thermal energy and produce a significant amount of heat around magnetic nanoparticles. The use of magnetic nanostructured supports can also allow for the convenient recovery of nanobiocatalysts [86]. Several interesting examples have successfully harnessed this heating effect in enzyme activation. For example, Knecht et al. prepared a Fe$_3$O$_4$ nanoparticle–enzyme hydrogel network and demonstrated that the thermophilic enzymes could be activated 2-fold via elevated temperature from magnetic actuation [26]. Notably, Xiong et al. developed a ferrimagnetic vortex-domain nanoring-enzyme hybrid and showed that the reaction rate of immobilized enzymes can be boosted up to 1.8-fold without heating up the solution and in a real-time manner (Figure 5B) [27]. This work, for the first time, provides direct biochemical evidence that the localized heating effect from remote magnetic stimulation can be utilized to specifically modulate enzymatic reactions, and may spark future research in spatiotemporally manipulation of intracellular catalysis in living organisms.

Localized heating of enzyme can be produced by the above-mentioned nanostructured supports when exposed to light or alternating magnetic field. The temperature increase at the nanoparticle surface with a sub-nanometer resolution was found to scale down linearly [87]. High local heating may lead to significant damage in enzyme conformation and activity. Hence, it is necessary to weigh the pros and cons of temperature effects on enzyme activity. It may be useful to adjust the distance between enzyme and support or use thermosensitive coupling molecules for enzyme binding.

*2.5. Conformational Changes of Immobilized Enzymes*

Favorable conformational changes of enzymes induced by either an allosteric effector or nanostructure supports can contribute to improved catalytic activity of nanobiocatalysts [88,89]. Some metal ions have been used to transform the conformation of enzymes to an active form by binding with certain amino acid residues outside enzyme active sites. The so-called allosteric effect is a striking example and a powerful tool to improve the activities of nanobiocatalysts [53]. For example, Zeng et al. reported CaHPO$_4$-$\alpha$-amylase nanoflowers which have dramatically (38-fold) increased enzyme activities [17]. In this system, Ca$^{2+}$ ions serve as effectors to bind at an allosteric site, leading to the activation of the enzyme molecules (Figure 5C). It was also found that Mg$^{2+}$ may act as an allosteric effector of β-galactosidase and was shown to be able to change the secondary structures of enzymes in β-galactosidase/Mg-Al layered double hydroxides [58]. It is clear that every allosteric enzyme is responsive to certain metal ions; consequently, there is no all-encompassing "one size fits all" solution to increase the activity of nanobiocatalysts. The allosteric effectors, apart from metal ions, can be modified on nanostructured supports, which may be a more general and highly effective way to activate allosteric enzymes. In addition, Li et al successfully immobilized lipase from pseudomonas cepacia on siliceous mesocelluar foams (MCF) with different hydrophobicity [71]. With the increase in the surface hydrophobicity of the MCFs, the catalytic activity of lipase was enhanced up to 25-fold. The lipase activation was attributed to hydrophobic interactions between the alkyl groups of MCF and the surface loops of enzymes to produce favorable conformational changes. In other examples, surface modifications of nanostructured supports have been shown to have significant influences not only on enzyme conformation but also on substrate binding in the nanobiocatalysts [12,64,66,90–92].

*2.6. Multi-Enzyme System*

A multi-enzyme system can be obtained by co-immobilizing two or more enzymes, which realize the micro- and nanoscale compartments and enhance the overall activity of the enzymes. There are many strategies for constructing multienzyme catalysis, such as co-immobilization of enzymes, conjugation of natural enzymes with artificial enzymes (i.e. nanozymes), and enhancing the function of enzymes within cell-free metabolic pathways [59,65,77,93–97]. Encapsulation is a common form to maintain a high local concentration of enzymes and protect them from biological damage through proteases. It was reported that five of six encapsulated enzymes inside a DNA nanocage exhibited increased activity and similar $K_m$ in comparison with free enzymes [98]. The negatively charged phosphate groups and highly structured water surface on the DNA surface may play a key role in stabilizing the active enzyme conformation.

The activity enhancement phenomenon of multienzyme systems has attracted increasing interest, especially in designing strategies to achieve favorable substrate channeling in cascade reactions [99,100]. Ge et al. and co-workers reported a spatially co-immobilized GOx–HRP system which exhibited dramatically enhanced overall catalytic performance [65]. In this case, the compartmentalization of the two enzymes endow the multi-step cascade reaction with ordered substrate transport, and the formed intermediate concentration gradients make a great contribution to driving and facilitating the reaction. Vranish et al. synthesized a multi-enzymatic coupled system by binding HRP and GOx on quantum dots and for the first time demonstrated a >2-fold improvement in the $k_{cat}$ of the system (Figure 5D) [59]. The challenge is, however, that not all enzymes will manifest activity enhancement when binding to the same quantum dots. Li et al. reported a bio-inspired nanozyme-based multienzyme system which consisted of Au nanoparticles with two enzyme-like activities and ATP synthase, by mimicking mitochondrial oxidative phosphorylation [97]. In this system, the intrinsic glucose oxidase and peroxidase activities of Au nanoparticles provided favorable proton gradient, driving the oxidative phosphorylation of ATP synthase as efficiently as natural mitochondria. In conclusion, nanozymes have been booming in recent years and providing a novel way for designing highly efficient multienzyme systems [60,101,102].

Multi-enzyme systems should be designed to minimize the diffusion of intermediates among the enzymes and increase their overall activity. It is expected that, to create multi-enzyme systems with synergic functions and spatiotemporal multicompartments [103], considerable attention and effort should be devoted to realizing control over the spatial arrangement, number, and class of the enzymes. Some new nanomaterials, such as metal–organic framework nanomaterials, Janus nanoparticles, mesocrystal nanozymes, and aggregation-induced emission nanoparticles, are considered ideal nanostructured supports to organize enzymes in confined microscale or nanoscale environments. In this way, the product of one enzyme can be channeled to act as substrate for the second enzyme. The obtained multi-enzyme system may enhance the overall activity based on high local substrate concentrations [104].

## 3. Conclusions

Nanobiocatalysts, as a result of the fusion of nanotechnology and biotechnology, exhibit remarkably increased activities, which is a breakthrough in the field of immobilized enzymes. We reviewed the recent development in design and application of these nanobiocatalysts, and summarized underlying the mechanisms of favorable interactions between the nanostructured supports and enzymes, including metal ion activation, enhanced electron transfer, morphological effects, conformational modulation, temperature effects and multi-enzyme systems. To further promote the applications of nanobiocatalysts, the following urgent challenges need to be addressed: (1) an evolutional nanobiocatalyst that can permit the simple recycle and reuse of enzymes; (2) minimizing the "dead areas" of nanobiocatalysts in catalytic reactions to make the hybrid systems more economically friendly; (3) improving the biocompatibility and stability of nanobiocatalysts for in vivo and in vitro biomedical applications; (4) smart nanobiocatalysts that can respond efficiently to remote stimuli for modulating the activities of nanobiocatalysts on demand. We firmly believed that the rapid development of nanotechnology will offer great opportunities to construct novel nanobiocatalysts with real-time-adjustable activity, with widespread applications in biology, medicine and environmental engineering.

**Author Contributions:** J.A. and G.L. wrote the manuscript and prepared the revised manuscript. Y.Z. and T.Z. discussed the ideas. X.L. and F.G. researched the literature and collected the literature reports. M.P. and Y.H. aided in writing the manuscript. H.F. coordinated the writing and outlined the manuscript. All authors have read and agreed to the published version of the manuscript.

**Funding:** This research received no external funding. The APC was funded by Northwest University.

**Acknowledgments:** This work was supported by the National Natural Science Foundation of China (No. 81771981, 81571809, and 31400663) and the Shaanxi Provincial Key Research and Development Program (2019KW-078).

**Conflicts of Interest:** The authors declare no conflict of interest.

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
