# Peer review of "Recent Advances in Enzyme-Nanostructure Biocatalysts with Enhanced Activity"

_catalysts, doi:10.3390/catal10030338_

Round 1
Reviewer 1 Report
In my opinion this article (after the revisions) can be accepted for a publication in Catalysts.
Author Response
Thanks for your kind suggestions!
Reviewer 2 Report
The manuscript has been significantly improved and it is ready for publication.
Author Response
Thanks for the kind suggestions!
Reviewer 3 Report
This review by Jing et al. is well written. The nanostructured biocatalysts have been described comprehensively. There are few concerns needed to be addressed by the authors.
1) Protein and Support structure inserts in Figure 1 does not convey the message properly. Figure 1 can be drawn as authors showed in their previous version of the manuscript without those insert elements.
2) I encourage authors to give or cite examples of each subset in Figure 2.
3) I see authors have summarized a table for the effect of metal ion and temperature in the enzyme activity following the suggestion from their previous submission.
Author Response
Point by point response
Comments to the Author
1. Protein and Support structure inserts in Figure 1 does not convey the message properly. Figure 1 can be drawn as authors showed in their previous version of the manuscript without those insert elements.
Reply:
Thank the reviewer for your kind comments. Figure 1 has been redrawn as the previous version in the revised manuscript.
2. I encourage authors to give or cite examples of each subset in Figure 2.
Reply:
Thanks a lot for your constructive suggestions. We have cite examples of each subset in Figure 2 as stated in the revised manuscript. “As shown in Figure 2, nanobiocatalysts with increased activities are constructed through insertion of nanoparticles into enzyme molecules [16], immobilization of enzymes on the surface of nanoparticle/nanorod/2 dimensional nanomaterial [21, 22, 32], encapsulation of enzymes in porous supports [35, 37], embedding/enveloping enzymes on/in 3 dimensional nanostructured supports [19, 42, 66].”
Figure 2. Representative morphologies of nanobiocatalysts with increased activities. (A) Insertion of nanoparticles into enzyme molecules. (B, C, D) Immobilization of enzymes on the surface of nanoparticle/nanorod/2 dimensional nanomaterial. (E) Encapsulation of enzymes in porous supports. (F, G, H) Embedding/enveloping enzymes on/in 3 dimensional nanostructured supports.

Reviewer 4 Report
The suggestions were partially improved.
Author Response
Thanks for your kind suggestions!